# A Deep Learning-Based Method for Overhead Contact System Component Recognition Using Mobile 2D LiDAR

**DOI:** 10.3390/s20082224

**Published:** 2020-04-15

**Authors:** Lipei Chen, Cheng Xu, Shuai Lin, Siqi Li, Xiaohan Tu

**Affiliations:** College of Computer Science and Electronic Engineering, Hunan University, Changsha 410082, China; lipeic@hnu.edu.cn (L.C.);

**Keywords:** deep learning, LiDAR, MLS, OCS, point cloud, railway

## Abstract

The overhead contact system (OCS) is a critical railway infrastructure for train power supply. Periodic inspections, aiming at acquiring the operational condition of the OCS and detecting problems, are necessary to guarantee the safety of railway operations. One of the OCS inspection means is to analyze data of point clouds collected by mobile 2D LiDAR. Recognizing OCS components from the collected point clouds is a critical task of the data analysis. However, the complex composition of OCS makes the task difficult. To solve the problem of recognizing multiple OCS components, we propose a new deep learning-based method to conduct semantic segmentation on the point cloud collected by mobile 2D LiDAR. Both online data processing and batch data processing are supported because our method is designed to classify points into meaningful categories of objects scan line by scan line. Local features are important for the success of point cloud semantic segmentation. Thus, we design an iterative point partitioning algorithm and a module named as Spatial Fusion Network, which are two critical components of our method for multi-scale local feature extraction. We evaluate our method on point clouds where sixteen categories of common OCS components have been manually labeled. Experimental results show that our method is effective in multiple object recognition since mean Intersection-over-Unions (mIoUs) of online data processing and batch data processing are, respectively, 96.12% and 97.17%.

## 1. Introduction

Rail transportation is a significant part of the transportation network. Up to 2019, the mileage of the global railway is over 1.3 million kilometers [1]. Billions of passengers and cargoes are transported by rail every year. The large-scale railway with high traffic challenges the stability and safety of the railway system. The overhead contact system (OCS) is a critical railway infrastructure for train power supply [2]. The components of OCS, such as contact wires, are easily deformed because of the contact force from the pantograph and various climatic conditions. The deformations might result in unstable power supply and even accidents. To access the operational condition of OCS and discover potential problems, periodic OCS inspections are required [3]. However, most of the OCS inspections nowadays are still done by workers. For instance, in China, workers with relative expertise and experience manually measure geometries of OCS components under the help of a portable laser rangefinder. With continuously growing mileage of railway [4], the manual measurement, which is inefficient and high-cost, might be hard to meet demands in the future.

Mobile laser scanning (MLS) is an emerging technology used for capturing dense three-dimensional (3D) point cloud. An MLS system with quick and accurate data acquisition can well record geometric details of surrounding objects [5]. Therefore, it is appropriate to apply MLS systems to OCS inspections. The operational condition of OCS can be acquired by analyzing the MLS point cloud instead of manual measurement. Recognizing the point cloud of OCS as a critical task of the data analysis has been studied in previous studies (e.g., [6,7,8]). Mobile 2D LiDAR is a special kind of MLS system applied to railway inspections. Figure 1 shows an instance of the mobile 2D LiDAR used to scan the OCS infrastructure. As for this kind of MLS system, a point cloud is built up by integrating points at each 2D scan line. Fully understanding the point cloud is significant for automatic inspections and intelligent diagnoses. Thus, this study focuses on multiple OCS component recognition with mobile 2D LiDAR. However, it is a difficult task because the similarities and the various associations among OCS components make scenes complex [8]. In this case, model-driven and data-driven methods become incompetent because rules and features for recognizing specific objects are difficult to be designed by human beings. Fortunately, the success of deep learning-based image recognition (e.g., [9,10,11]) has promoted the fast development of deep learning-based point cloud segmentation, which provides novel means to understand point cloud with semantics.

In this study, we propose a new deep learning-based method to conduct semantic segmentation on the point cloud collected by mobile 2D LiDAR. Points are classified into meaningful categories of objects scan line by scan line by our method. As local features are important for the success of point cloud semantic segmentation [12,13], an iterative point partitioning algorithm is developed to partition points into regions for local feature extraction at each scan line, and the Recurrent Neural Network (RNN)-based module, named as Spatial Fusion Network (SFN), is developed to extract and fuse local features among scan lines. We leverage the iterative point partitioning algorithm and SFN to perform multi-scale feature extraction, which can improve the segmentation performance. Experiments are carried out on the MLS point cloud collected from the OCS of Chinese high-speed railways. Experimental results show that our method is light and fast to classify points with high accuracy. To verify the effectiveness of our design, we further analyze the influences of different settings of feature extraction and feature fusion. The contributions of our work are summed up as follows:A new deep learning-based method is proposed to solve the problem of recognizing multiple OCS components from the point cloud collected by mobile 2D LiDAR. Experiments show that our method is effective in multiple object recognition.To solve the issue of local feature extraction, we develop an iterative point partitioning algorithm and SFN to acquire local features among scan lines at multiple scales.Our method is designed to classify points scan line by scan line, supporting both online data processing and batch data processing.

The rest of this paper is organized as follows. Section 2 presents the previous works of railway object recognition based on MLS data, some existing commercial solutions of OCS inspections, and some deep learning-based architectures of point cloud semantic segmentation. Section 3 illustrates the methodology. Section 4 describes the experiments and experimental results. Section 5 draws conclusions and the vision of the future.

## 2. Related Work

In the previous studies, some methods have been proposed to recognize railway elements from MLS data. These methods can be mainly categorized into two types: model-driven and data-driven methods and learning-based methods.

With model-driven and data-driven methods, points are grouped into non-overlap regions based on manually designed features as well as handcrafted rules. For instance, Arastounia et al. [14,15,16] recognized the railway infrastructure based on templates, local structures, shapes of objects and topological relationships among objects. Lou et al. [17] extracted railway tracks from MLS data based on height difference calculated by using a sliding window method. Stein et al. [18] detected rails and tracks in 2D LiDAR data based on the extracted geometrical and topological features, and a rail shape template. Zou et al. [19] leveraged vector shift estimation, K-mean clustering, and reverse smoothing to extract rail tracks, resolving the difficulties of rail track extraction in scenarios of bends and turnout. Pastucha [6] recognized objects of OCS by using the RANSAC algorithm and DBSCAN algorithm. Gézero et al. [20] designed a simple algorithm to extract the linear elements of the railway based on the assumption that the extracted elements are roughly parallel to the sensor system trajectory. However, manually designed features and handcrafted rules are often designed for recognizing specific objects and difficult to extend to new tasks. Besides, a rule for recognizing various objects from a point cloud is difficult to design by human beings because it might be very complicated.

Compared to model-driven and data-driven methods, learning-based methods are more promising to fully understand the point cloud of complex scenes because features are automatically learned from data rather than manually designed. The regular supervised machine learning approach has been introduced to recognize railway objects. For instance, Jung et al. recognized ten categories of railway electrification system objects from the MLS point cloud by using a multi-range CRF model [7] or a multi-scale hierarchical CRF model [8]. However, few studies introduced the deep learning approach to recognized railway objects from MLS data. In contrast, some deep learning-based methods [21,22,23,24,25,26,27,28,29] have been successfully applied to image-based OCS inspections for component detection and defect detection.

As for commercial solutions of OCS inspections, the image processing technology and the MLS technology are commonly introduced. Due to the mature development of image processing technology, some methods, such as edge detection, model-matching, and deep learning-based image recognition, are used for defect detection. For instance, JX-300 [30] is a kind of inspection vehicle equipped with an image-based OCS inspection system. Compared with image data, MLS data contain 3D information, which is useful to measure geometric parameters of the OCS components. For instance, tCat [31] is a real-time mobile mapping solution using a portable rail trolley with laser sensors to measure geometries of overhead wires. Selectra Vision Company [32] developed LiDAR-based equipment for contact wire measurement. In addition, some solutions combined the image processing technology and the MLS technology to build up OCS inspection systems. For instance, Meidensha Corporation [33,34] combined laser scanners and line scan cameras to detect the position of contact wires. MERMEC Company [35,36] developed an inspection system that is made up of lasers and cameras to monitor the overhead wires and pick up defects.

Nowadays, deep learning-based methods of point cloud semantic segmentation provide novel means to recognize various objects in point cloud scenes. PointNet [37] proposed by Qi et al. is a pioneer of deep learning-based architecture in the field of 3D point cloud classification and semantic segmentation. However, the lack of local information restricts the performance of PointNet in large and complicated scenarios. Then, Qi et al. further proposed PointNet ++ [12] to overcome disadvantages of PointNet by using a hierarchical structure to capture local features of a point cloud, which achieves strong robustness and good effect. Since local features are significant to improve the performance of 3D point cloud semantic segmentation, some studies have been conducted for local feature extraction. Ye et al. [38] proposed a model named as 3P-RNN, in which local features are extracted by using Pointwise Pyramid Pooling modules and RNNs. Some architectures extract local features based on convolutional operations, such as PointCNN [39], Pointwise CNN [40], DGCNN [41], and LDGCNN [13].

## 3. Methodology

As for the long-distance OCS inspection, the MLS system would collect millions of points. It is a tremendous task to compute point relations in the whole point cloud. To reduce computational complexity, we process data scan line by scan line, avoiding computing relations between points of different scan lines.

The framework of our method is shown in Figure 2. As for feature extraction, we are concerned about two issues: (1) we have to generate regions for local feature extraction; and (2) 3D contexts are in demand for classifying points at a scan line because the 2D features extracted at a scan line are incompetent for recognizing 3D objects. To solve these two issues, we leverage the iterative point partitioning algorithm to generate regions and SFN to acquire 3D contexts by feature extraction and feature fusion. Details of the proposed framework are described in the following sections.

### 3.1. Data Preprocessing

In the 2D polar coordinate system, the *t*th scan line can be represented as a set of 2D points as:(1)$$A_{t}=\left\{\left(\alpha_{t,i},\theta_{t,i}\right)|i=1,\ldots ,N\right\}$$
where $\alpha_{t,i}$ is a range distance, $\theta_{t,i}$ is a bearing angle, and *N* is the number of points. Combined with traveled distances, we can transform points of scan lines into the 3D Cartesian coordinate system defined in Figure 3. Let $d_{t}$ be the traveled distance of the *t*th scan line, and let $B_{t}$ be a set of the 3D points transformed from $A_{t}$. Let *u* and *v* be, respectively, the horizontal offset and vertical offset of the LiDAR based on the rail track centerline. The point set $B_{t}$ can be written as:(2)$$B_{t}=\left\{p_{t,i}\left(x_{t,i},y_{t,i},z_{t,i}\right)|i=1,\ldots ,N\right\}$$
where $x_{t,i}$, $y_{t,i}$, and $z_{t,i}$ are coordinates computed as follows:(3)$$\left[\begin{array}{c} x_{t,i}\\ y_{t,i}\\ z_{t,i} \end{array}\right]=\alpha_{t,i}\left[\begin{array}{c} -\cos\theta_{t,i}\\ \sin\theta_{t,i}\\ 0 \end{array}\right]+\left[\begin{array}{c} u\\ v\\ d_{t} \end{array}\right]$$
To reduce computation, we set up an inspection area to filter out distant points that belong to background objects such as trees and buildings. Let $P_{t}$ be a set of points within the inspection area. It can be written as:(4)$$P_{t}=\left\{p_{t,i}\left(x_{t,i},y_{t,i},z_{t,i}\right)\in B_{t}|x_{min}<x_{t,i}<x_{max},y_{min}<y_{t,i}<y_{max}\right\}$$
where $x_{min}$, $x_{max}$, $y_{min}$, and $y_{max}$ describe the inspection area. By integrating the points within the inspection area, we can form a point cloud for OCS inspection. The point cloud Pc can be represented as:(5)$$Pc={⋃}_{t=1}^{T}P_{t}$$
where *T* is the number of scan lines.

### 3.2. Iterative Point Partitioning

The proposed algorithm for partitioning points is conducted iteratively at each scan line to generate regions and establish region-to-region relationships. Note that a region is a set of points in a local neighborhood. By this algorithm, points are partitioned based on the region centroids acquired previously. As defined in Figure 4, region size and reign span are concepts applied to the algorithm:To extract local features at a specified scale, we group points into non-overlap regions with the region size which is limited by a region size threshold ε.To build up the region-to-region relationship in a 3D local neighborhood for feature fusion, the region span between two regions has to be less than the span threshold δ. The value of δ is based on human experiences of OCS object shapes.

At the *t*th scan line, points within the inspection area can be written as:(6)$$P_{t}=\left\{p_{t,i}\left(x_{t,i},y_{t,i},z_{t,i}\right)|i=1,\ldots ,N_{t}\right\}$$
where $N_{t}$ is the number of points and $z_{t,1}=z_{t,2}=\ldots =z_{t,N_{t}}=d_{t}$ denotes the traveled distance of the scan line. Let $C_{t-1}^{\varepsilon }=\left({\hat{c}}_{t-1,1}^{\varepsilon },\ldots ,{\hat{c}}_{t-1,K}^{\varepsilon }\right)$ be the list of the *K* region centroids acquired previously with the region size threshold ε. The specific steps of our iterative point partitioning algorithm are as follows:To restrict the region span within δ, we filter the region centroids in $C_{t-1}^{\varepsilon }$ by discarding the ones that do not satisfy the following condition:
(7)$$\left|d_{t}-{\hat{z}}_{t-1,n}^{\varepsilon }\right|<\delta$$
where ${\hat{z}}_{t-1,n}^{\varepsilon }$ denotes the z-coordinate value of ${\hat{c}}_{t-1,n}^{\varepsilon }$.Points in $P_{t}$ are matched to the remaining region centroids by the nearest matching strategy. The nearest matching strategy is based on the match function defined as follows:
(8)$$match\left(p,C\right)=\underset{\hat{c}\in C}{\arg\min}Dist3D\left(p,\hat{c}\right)$$
where *p* is a point to be matched, *C* is a set of region centroids, and Dist3D function computes the distance between *p* and $\hat{c}$ in the 3D coordinate system.To restrict the region size within ε, we discard the matching relations that do not satisfy the following condition:
(9)$$Dist2D\left(p_{t,i},{\hat{c}}_{t-1,n}^{\varepsilon }\right)<\varepsilon /2,\;\mathrm{where}\;{\hat{c}}_{t-1,n}^{\varepsilon }=match\left(p_{t,i},C_{t-1}^{\varepsilon }\right)$$
where Dist2D function computes the distance between $p_{t,i}$ and ${\hat{c}}_{t-1,n}^{\varepsilon }$ in the xy-coordinate system, and the match function is defined in Step 2. The points without matching relation are grouped into an unmatched point set.Assume that ${\hat{c}}_{t-1,n}^{\varepsilon }$ is the region centroid of region $G_{t-1,n}^{\varepsilon }$ (as described in Step 6, ${\hat{c}}_{t-1,n}^{\varepsilon }$ might be the region centroid of another previous region which is not empty). Points matched to ${\hat{c}}_{t-1,n}^{\varepsilon }$ are positionally near to $G_{t-1,n}^{\varepsilon }$ so that they can be grouped together to form a new region $G_{t,n}^{\varepsilon }$ for feature fusion. Then, the region-to-region relationship between $G_{t-1,n}^{\varepsilon }$ and $G_{t,n}^{\varepsilon }$ can be built up. Note that *n* of the symbols formatted as $G_{t,n}^{\varepsilon }$ denotes region-to-region relationships. Based on Equations (7) and (9), the new region $G_{t,n}^{\varepsilon }$ can be written as:
(10)$$G_{t,n}^{\varepsilon }=\left\{p_{t,i}\in P_{t}|\left|d_{t}-{\hat{z}}_{t-1,n}^{\varepsilon }\right|<\delta ,{\hat{c}}_{t-1,n}^{\varepsilon }=match\left(p_{t,i},C_{t-1}^{\varepsilon }\right),\;Dist2D\left(p_{t,i},{\hat{c}}_{t-1,n}^{\varepsilon }\right)<\varepsilon /2\right\}$$The hierarchical clustering algorithm is utilized to partition the unmatched points which have been acquired in Step 3. The furthest-neighbor strategy [42] with ε as the merging threshold is applied to this process.The regions generated with ε can be represented as $G_{t,1}^{\varepsilon },\ldots ,G_{t,K+M}^{\varepsilon }$, where $G_{t,1}^{\varepsilon },\ldots ,G_{t,K}^{\varepsilon }$ are the regions acquired in Step 4 by matching, and $G_{t,K+1}^{\varepsilon },\ldots ,G_{t,K+M}^{\varepsilon }$ are the regions acquired in Step 5 by clustering. Based on these regions, we produce the region centroid list $C_{t}^{\varepsilon }=\left({\hat{c}}_{t,1}^{\varepsilon },\ldots ,{\hat{c}}_{t,K+M}^{\varepsilon }\right)$ for the next iterative process, where:
(11)$${\hat{c}}_{t,n}^{\varepsilon }=\left\{\begin{array}{cc} {\hat{c}}_{t-1,n}^{\varepsilon } & ,G_{t,n}^{\varepsilon }=\emptyset \\ \frac{\sum_{p\in G_{t,n}^{\varepsilon }}p}{\left|G_{t,n}^{\varepsilon }\right|} & ,G_{t,n}^{\varepsilon }\neq \emptyset \end{array}\right.$$To capture point-to-point relations and region-to-region relations, we transform points into the local 2D coordinate system relative to their region centroids, and region spans between regions whose features would be fused are computed:
(12)$$\begin{array}{c} U_{t,n}^{\varepsilon }=\left\{q\left(\frac{x-{\hat{x}}_{t,n}^{\varepsilon }}{\varepsilon /2},\frac{y-{\hat{y}}_{t,n}^{\varepsilon }}{\varepsilon /2}\right)|p\left(x,y,z\right)\in G_{t,n}^{\varepsilon }\right\}\\ s_{t,n}^{\varepsilon }=\left\{\begin{array}{cc} \frac{d_{t}-{\hat{z}}_{t-1,n}^{\varepsilon }}{\delta } & ,n\leq K\\ 0 & ,n>K \end{array}\right. \end{array}$$
where n=1,…,K+M; $U_{t,n}^{\varepsilon }$ is a set of 2D points in the local coordinate system; $s_{t,n}^{\varepsilon }$ is a region span; ${\hat{x}}_{t,n}^{\varepsilon }$ and ${\hat{y}}_{t,n}^{\varepsilon }$ are, respectively, the x-coordinate value and the y-coordinate value of ${\hat{c}}_{t,n}^{\varepsilon }$; and ε and δ are utilized to scale the value into $\left[-1,1\right]$.

### 3.3. Spatial Fusion Network

As PointNet is an effective architecture to extract features from unordered points [37], a PointNet layer is introduced to SFN for feature extraction. In addition, impelled by the success of RNN in video recognition [43,44,45,46], we utilize RNNs to fuse features among different scan lines to acquire 3D contexts. Specifically, RNNs in our SFN are implemented as two single-layer Long Short-Term Memory (LSTM) networks [47]. The architecture of SFN is shown in Figure 5. The inputs, represented as a set of 2D points in their local coordinate system as well as a value of region span, are produced by the iterative point partitioning algorithm (Step 7). At the *t*th iteration, the input points can be represented as $\left\{q_{t,1},\ldots ,q_{t,M_{t}}\right\}$, where $M_{t}$ is the number of input points, and the region span can be represented as $s_{t}$. The *t*th iterative process of SFN is described below in detail.

#### 3.3.1. PointNet Layer for Feature Extraction

In the PointNet layer, the input points are firstly mapped to a high-dimensional feature space by a multilayer perceptron (MLP):(13)$${fq}_{t,i}=MLP\left(q_{t,i}\right),\;\mathrm{for}\;i=1,\ldots ,M_{t}$$
where $MLP\;:\;R^{2}\to R^{128}$, and ${fq}_{t,i}$ is the point feature of $q_{t,i}$. Then, the point features are aggregated by a max-pooling layer to form a region feature:(14)$${fr}_{t}=\max\left({fq}_{t,1},\ldots ,{fq}_{t,M_{t}}\right)$$
where ${fr}_{t}$ is the region feature aware of the relation of input points.

#### 3.3.2. Recurrent Neural Networks for Feature Fusion

For feature fusion, the input region span is firstly used to produce the span feature which represents the spatial relationships between regions. Then, the span feature is concatenated with the region feature together for feature fusion. There are two optional kinds of LSTM:unidirectional LSTM (UDLSTM), which uses forward dependencies to fuses features; andbidirectional LSTM (BDLSTM), which make use of the forward and backward dependencies to form complete 3D perspectives for feature fusion.
As for using the UDLSTM, the process of feature fusion can be written as follows:(15)$$\begin{array}{c} {fs}_{t},h_{t}^{1}={LSTM}^{1}\left(s_{t},h_{t-1}^{1}\right)\\ {ff}_{t},h_{t}^{2}={LSTM}^{2}\left(concat\left({fr}_{t},{fs}_{t}\right),h_{t-1}^{2}\right) \end{array}$$
and the process with the usage of the BDLSTM can be written as follows:(16)$$\begin{array}{c} {fs}_{t},h_{t}^{1}={LSTM}^{1}\left(s_{t},h_{t-1}^{1}\;,h_{t+1}^{1}\right)\\ {ff}_{t},h_{t}^{2}={LSTM}^{2}\left(concat\left({fr}_{t},{fs}_{t}\right),h_{t-1}^{2},h_{t+1}^{2}\right) \end{array}$$
where ${fs}_{t}$ is the span feature; ${fr}_{t}$ is the region feature acquired in Equation (14); ${ff}_{t}$ is the new region feature with a 3D context; $h_{t-1}^{1}$, $h_{t}^{1}$, and $h_{t+1}^{1}$ are hidden states of ${LSTM}^{1}$; $h_{t-1}^{2}$, $h_{t}^{2}$, and $h_{t+1}^{2}$ are hidden states of ${LSTM}^{2}$; and concat function is used to concatenate the input features.

By combining the point feature and the region feature, we can capture relationships between points in a 2D region. By feature fusion, we can acquire contexts in 3D neighborhoods. Therefore, we concatenate ${fr}_{t}$ and ${ff}_{t}$ with each ${fq}_{t,i}$, to form the new point features with 3D contexts:(17)$${fp}_{t,i}=concat\left({fq}_{t,i},{fr}_{t},{ff}_{t}\right),\;\mathrm{for}\;i=1,\ldots ,M_{t}$$
where ${fp}_{t,i}$ is the new point features of $q_{t,i}$.

### 3.4. Multi-Scale Feature Extraction and Per-Point Classification

Figure 6 shows an instance of single-scale feature extraction. As for multi-scale feature extraction, we conduct the single-scale feature extraction repeatedly with different region size thresholds. At the *t*th scan line, point features extracted at *K* scales can be written as:(18)$${Fp}_{t}^{\varepsilon_{k}}=\left\{{fp}_{t,1}^{\varepsilon_{k}},\ldots ,{fp}_{t,N_{t}}^{\varepsilon_{k}}\right\},\;\;k=1,\ldots .,K$$
where ${fp}_{t,i}^{\varepsilon_{k}}$ extracted with the region size threshold $\varepsilon_{k}$ is a point feature of $p_{t,i}$ and $N_{t}$ is the number of points within the inspection area at the *t*th scan line. Point features extracted at multiple scales are concatenated together to form multi-scale point features. Then, the multi-scale point features are input to an MLP to yield point scores:(19)$$\begin{array}{c} {fp}_{t,i}=concat\left({fp}_{t,i}^{\varepsilon_{1}},\ldots ,{fp}_{t,i}^{\varepsilon_{K}}\right)\\ {score}_{t,i},=MLP\left({fp}_{t,i}\right) \end{array},\;\mathrm{for}\;i=1,\ldots .,N_{t}$$
where $fp_{t,i}$ denotes the multi-scale point feature of $p_{t,i}$; ${score}_{t,i}$ is the point sore indicating which class $p_{t,i}$ should belong to. In Equation (19), MLP with $\left(512,\;128,\;32,\;c\right)$ as its four hidden layer sizes are shared for yielding point scores at each scan line, where *c* is the number of categories of targets. Additionally, batch normalization is applied to each layer of the MLP with ReLU.

## 4. Experiments

### 4.1. Data Description

The MLS data for training and testing were collected from parts of the Chinese high-speed railway with a length of 16.7 km in 2019. The SICK LMS511 LiDAR sensor is applied to our MLS system for data acquisition. Table 1 shows the operating parameters of our MLS system, and Figure 7a shows the scene where data were acquired.

Based on the design of OCS, the inspection area described in Section 3.1 is set as $x_{min}=-1$ m, $x_{max}=7$ m, $y_{min}=0$ m, and $y_{max}=8$ m. Accordingly, there are 7.86 million points collected in this inspection area. We manually labeled the collected points into 17 classes: contact wire (COW), catenary wire (CAW), cantilever (CTLV), dropper (DRO), stitch wire (SW), pole, spreader (SPD), return wire (RW), feeder wire (FW), suspension insulator (SI), pulley block (PB), tensioning weight (TW), guy wire (GW), mid-point anchor wire (MPAW), mid-point anchor auxiliary wire (MPAAW), electric connector (EC), and others. Figure 7b–d shows examples of the manually labeled point cloud. Approximately 80% of the data are used for training and the rest for testing. MLS data of the training dataset and the test dataset are sliced into sixty samples and five samples, respectively. The compositions of these two datasets are shown in Table 2.

### 4.2. Implementation Details

For multi-scale feature extraction, we extract features with three region size thresholds, namely $\varepsilon_{1}=0.5$ m, $\varepsilon_{2}=2$ m, and $\varepsilon_{3}=7$ m, which are used for capturing the object relations at small scale, middle scale, and large scale. In addition, the region span threshold δ is set as 1 m. Figure 8 shows the results of iterative point partitioning with these three region size thresholds. As for feature fusion, UDLSTMs and BDLSTMs are, respectively, used in the SFN to evaluate our methods. Note that the mode with UDLSTMs is named as the unidirectional mode, and the mode with BDLSTMs is named as the bidirectional mode.

In terms of optimizing learning-based modules, we leverage the ten-fold cross-validation method on the training dataset to tune hyperparameters. Experimentally, we choose the cross-entropy loss function to estimate the loss. In addition, the Adam optimizer with the learning rate 0.001 is utilized to optimize parameters of the modules. The layer sizes and numbers of layers, described in Section 3.3 and Section 3.4, are determined by considering the segmentation accuracy and the computational efficiency. As the partitioning results are different among samples, the batch size is set as 1. Note that the batch size can be larger than 1 with padding alternatively, whereas it will bring in extra computation.

### 4.3. Semantic Segmentation Results

To evaluate our method, the evaluation metrics applied to our experiments are as follows:(20)$$precision=\frac{TP}{TP+FP}$$
(21)$$recall=\frac{TP}{TP+FN}$$
(22)$$IoU=\frac{TP}{TP+FP+FN}$$
(23)$$Accuarcy=\frac{TP+TN}{TP+FP+TN+FN}$$
where IoU denotes the Intersection-over-Union. TP, FP, TN and FN are, respectively, the numbers of true positives, false positives, true negatives, and false negatives.

Table 3 shows the point cloud semantic segmentation results on the test dataset. In the unidirectional mode and the bidirectional mode, high rates of mean precision (97.81% and 98.69%), mean recall (98.20% and 98.41%) and mean IoU (96.12% and 97.17%) are acquired by our method. By comparing the overall results of two modes, we can find that the more comprehensive 3D contexts provided in bidirectional mode can lead to a better segmentation performance. Major wires (i.e., contact wire, catenary wire, return wire, and feeder wire) are critical components for power transmission. Except for the catenary wire, the major wires are well recognized with IoU over 99% in both modes. Except for the dropper, stitch wire, and electric connector, the IoU of each class is over 94% in the unidirectional mode and over 96% in the bidirectional mode. The precision of stitch wire (90.37%) and electric connector (96.77%) are, respectively, the lowest in the unidirectional mode and the bidirectional mode. Besides, the electric connector class takes the lowest recall (90.59% and 90.90%) and the lowest IoU (86.75% and 88.23%) in both modes. The relatively low performance of recognizing electric connector objects might be caused by the small quantity of training samples.

As shown in Table 4 and Table 5, most of the errors of dropper are misclassifying dropper points as contact wire, catenary wire, and stitch wire. A few points of stitch wire and catenary wire are confused. In the unidirectional mode, a few electric connector points are incorrectly recognized as points of dropper and cantilever, whereas, in the bidirectional mode, points of electric connector are misclassified as dropper but not cantilever. In general, the errors shown in these two confusion matrices lead to the relatively low IoU of dropper, stitch wire, and electric connector.

Segmentation results are visualized in Figure 9 and Figure 10. Overall, few errors occur in the non-insulated overlap section. Compared with the non-insulated overlap section, the scene of the insulated overlap section is more complex because there are more relationships of interconnected objects. Nevertheless, objects are also clearly distinguished in the insulated overlap section. By highlighting errors, we can find that most of the errors occur at the intersections between objects.

### 4.4. Space and Time Complexity

We recorded the space and time cost using Pytorch 1.2 on a server with an Intel Xeon E5-2623 v4 CPU running at 2.6 GHz and an Nvidia Quadro M4000 GPU. The results are shown in Table 6. Compared with UDLSTM, using BDLSTM leads to more model parameters. As a result, our method in the bidirectional mode is more expensive than that in the unidirectional mode. The SLSL unidirectional mode consumes the most time because the iterative process cannot make full use of the GPU. The time cost without GPU (9.32 ms) just increases by approximately 3.67% in SLSL unidirectional mode. This result indicates that our method can be utilized to process scan lines with at least 100 Hz with Intel Xeon E5-2623 v4 CPU. Compared with the scanning frequency (25 Hz) of the LiDAR sensor used in the experiments, our method supports online data processing where data can be processed in real-time.

### 4.5. Ablation Study

Further experiments were conducted to explore the contributions of the multi-scale feature extraction and feature fusion.

To verify the effectiveness of multi-scale feature extraction, we compared the performances of different settings of region size threshold ε on the test dataset. As shown in Table 7, multiple-scale feature extraction contributes to remarkable improvements of overall accuracy (OA) and mIoU. Compared to the double-scale feature extraction (ε=0.5,7) in the unidirectional mode, the triple-scale feature extraction (ε=0.5,2,7) helps mIoU increase by 1.02%, although OA rises by 0.19%. The greater improvement of mIoU indicates that multiple-scale feature extraction can reduce errors of small-quantity classes.

To verify the effectiveness of feature fusion, we conducted an experiment in the non-feature fusion mode where we removed the feature fusion part of SFN, so that point features were only combined with region features. As depicted in Table 8, we can find that the rates of mIoU and OA increase with the help of feature fusion. In the non-feature fusion mode, features at a scan line only contain the 2D information, which would lead to confusion of recognizing 3D objects. For instance, points of a spreader are always misclassified as points of a cantilever at pole-spreader junctions in the non-feature fusion mode (Figure 11a), whereas few errors occur at that location in the unidirectional mode and the bidirectional mode (Figure 11b,c), which demonstrates that the feature fusion contributes to the reduction of errors.

## 5. Conclusions

This study focuses on multiple OCS component recognition with mobile 2D LiDAR. A deep learning-based method is developed to conduct point cloud semantic segmentation scan line by scan line. We propose the iterative point partitioning algorithm and SFN as two critical components of our method for local feature extraction. As shown in the experiments, sixteen categories of common OCS components are recognized with high precision, recall, and IoU on the test dataset. According to the design of SFN, UDLSTM and BDLSTM are two optional kinds of LSTM used for feature fusion, resulting in 96.12% and 97.17% mIoU, respectively. In particular, UDLSTMs with the mere forward dependencies allow the SLSL unidirectional mode to support online data processing. Additionally, the ablation study shows that the design of multi-scale local feature extraction makes remarkable improvements of the segmentation performance. Compared with the model-driven and data-driven methods mentioned in Section 2, our method with automatic feature learning rather than manually designed rules is more effective in multiple object recognition and easier to be extended to new recognition tasks.

With our further research, we intend to improve the iterative point partitioning algorithm and SFN to adapt to 3D input. Hence, our method can be applied to higher performance MLS systems equipped with 3D LiDAR. Furthermore, due to the ability of automatically learning features, it is possible to extend our method to other inspection tasks such as rail track inspections and tunnel inspections.

## Figures and Tables

**Figure 1 sensors-20-02224-f001:**
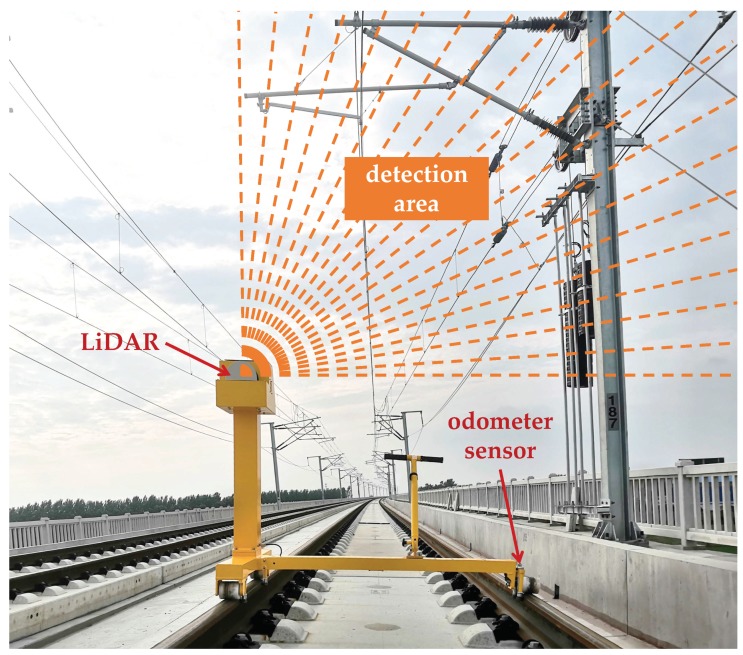
An instance of mobile 2D LiDAR used for data acquisition in this study. During the inspection, the 2D LiDAR scans the OCS infrastructure on the plane perpendicular to the rail direction, and the odometer sensor records the traveled distance of each scan line.

**Figure 2 sensors-20-02224-f002:**
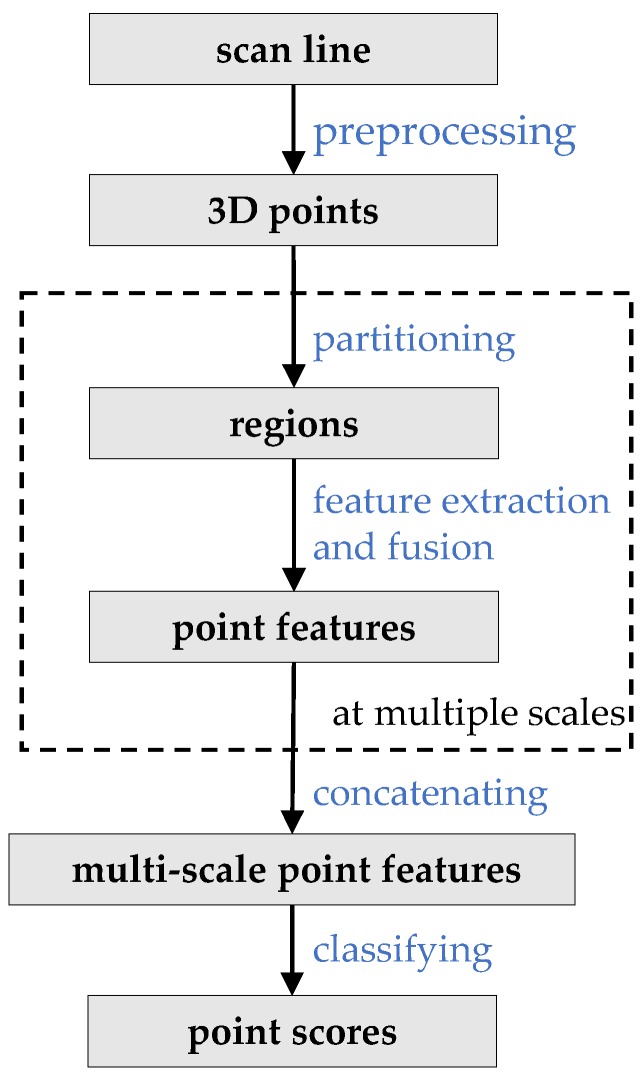
Data processing framework. Points are classified scan line by scan line. Firstly, data of a scan line are transformed into a 3D coordinate system. Secondly, point features with 3D contexts are extracted at multiple scales by the iterative point partitioning algorithm and the SFN. Finally, the multi-scale local features are concatenated together and then input to a classifier to yield point scores.

**Figure 3 sensors-20-02224-f003:**
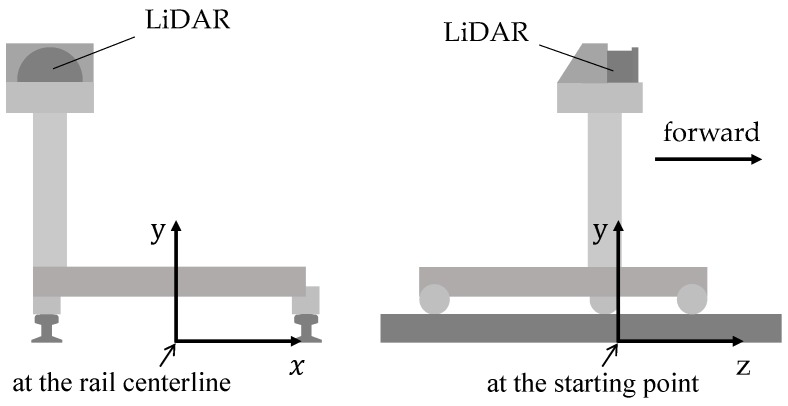
An MLS system carried by an inspection device at the starting point. The front view is shown on the left, and the left view is shown on the right. Since the device moves on the rail, the defined 3D Cartesian coordinate system is relative to rail tracks.

**Figure 4 sensors-20-02224-f004:**
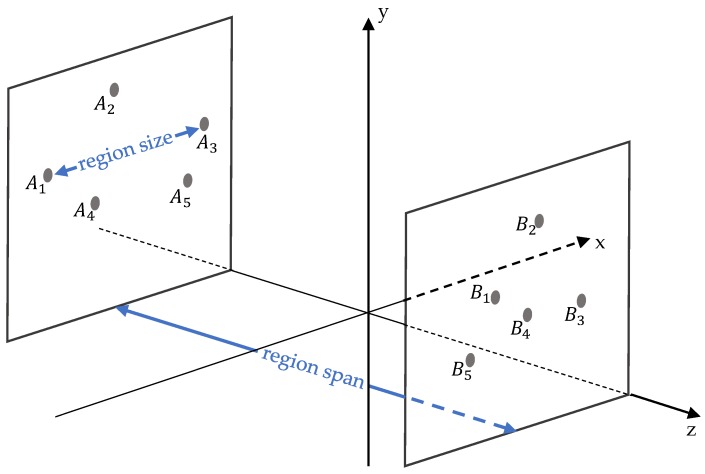
Definitions of region size and region span. Assume that there are region *A* and region *B*, respectively, in plane *A* and plane *B* which are perpendicular to the z-axis. In region *A*, $A_{1}$ and $A_{3}$ are the remotest pair of points. The distance between $A_{1}$ and $A_{3}$ denotes the region size of region *A*. The distance between plane *A* and plane *B* denotes the region span between region *A* and region *B*.

**Figure 5 sensors-20-02224-f005:**
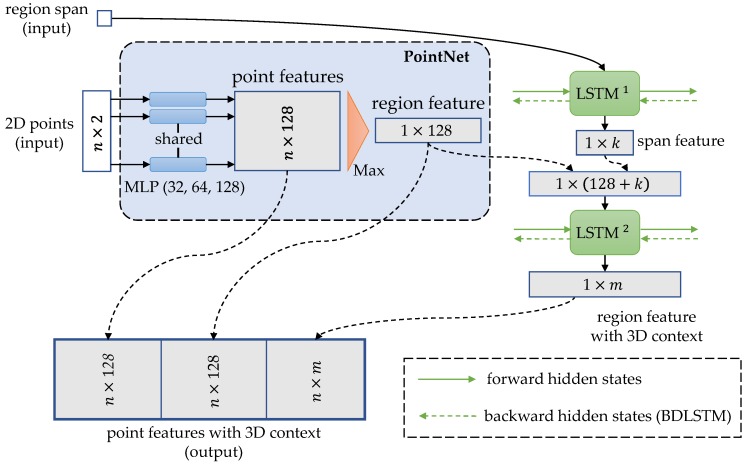
The architecture of SFN. In the PointNet, MLP denotes a multilayer perceptron, and batch normalization is applied to each layer of the MLP with ReLU. For feature fusion, both LSMTs are single hidden-layer. The hidden layer sizes of the LSTM${}^{1}$ and the LSTM${}^{2}$ are, respectively, 8 and 128 so that *k* and *m* are, respectively, 8 and 128 with the usage of UDLSTMs. Note that UDLSTMs are without backward hidden states. With the usage of BDLSTMs, outputs of the forward process and backward process are combined so that *k* and *m* are, respectively, 16 and 512.

**Figure 6 sensors-20-02224-f006:**
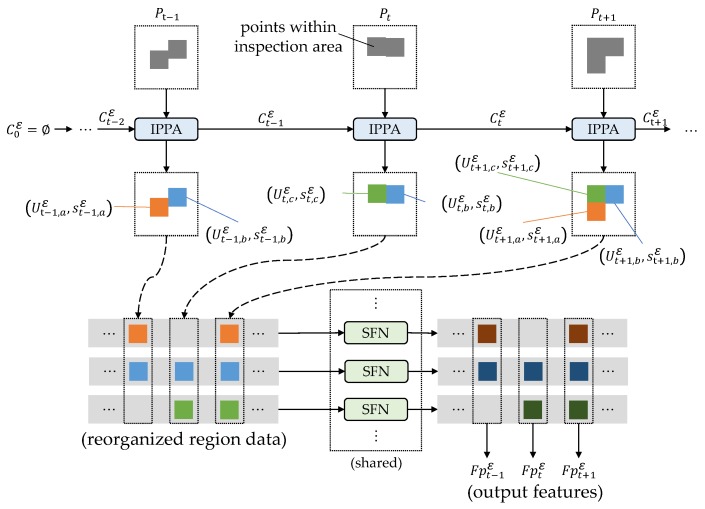
An instance of single-scale feature extraction with the region size threshold ε. Symbols formatted as $P_{t}$, $C_{t}^{\varepsilon }$, $U_{t,n}^{\varepsilon }$, and $s_{t,n}^{\varepsilon }$ are depicted in Section 3.2. At the beginning of the iterative process of point partitioning, $C_{0}^{\varepsilon }$ is initialized as ∅. Then, regions and region-to-region relationships are produced by the iterative point partitioning algorithm (IPPA). The region-to-region relationships are informed by the superscript *n* of the region data, which are formatted as $\left(U_{t,n}^{\varepsilon },s_{t,n}^{\varepsilon }\right)$. Note that the regions without points (e.g., region data $\left(U_{t,a}^{\varepsilon }=\emptyset ,s_{t,a}^{\varepsilon }\right)$) are discarded for feature extraction. Based on the region-to-region relationships, region data are reorganized in sequences (e.g., sequence $\left(\ldots ,\left(U_{t-1,a}^{\varepsilon },s_{t-1,a}^{\varepsilon }\right),\left(U_{t+1,a}^{\varepsilon },s_{t+1,a}^{\varepsilon }\right),\ldots \right)$). SFNs extract and fuse features among the sequential region data to acquire the point features with 3D contexts. We share the weights of SFNs for processing data at a single scale. The output ${Fp}_{t}^{\varepsilon }=\left\{{fp}_{t,1}^{\varepsilon },\ldots ,{fp}_{t,N_{t}}^{\varepsilon }\right\}$ is the set of point features corresponding to the input points $P_{t}=\left\{p_{t,1},\ldots ,p_{t,N_{t}}\right\}$.

**Figure 7 sensors-20-02224-f007:**
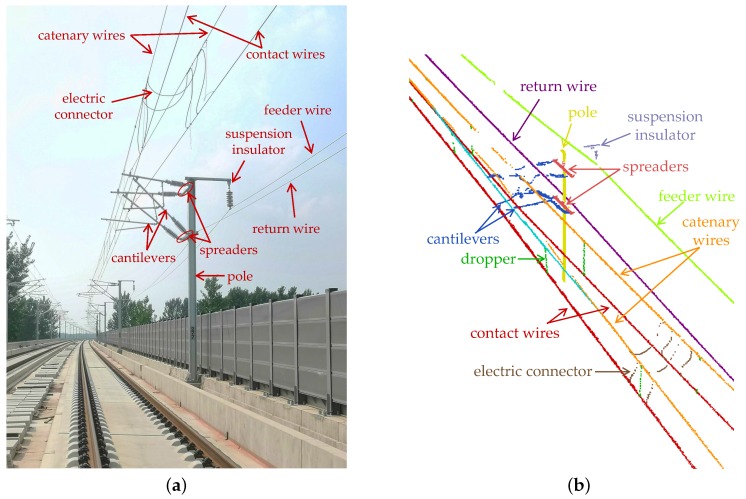
Overhead Contact System configuration of Chinese high-speed railway tagged with sixteen classes: (**a**) an image; and (**b**–**d**) MLS data ((**b**,**c**) sections of insulated overlap and (**d**) a section with mid-point anchor).

**Figure 8 sensors-20-02224-f008:**
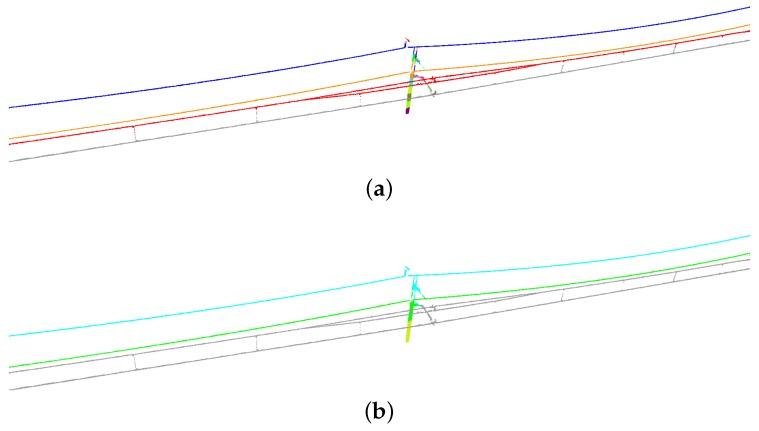
Results of iterative point partitioning: (**a**) $\varepsilon_{1}$ = 0.5 m; (**b**) $\varepsilon_{2}$ = 2 m; and (**c**) $\varepsilon_{3}$ = 7 m. The points grouped into regions are tagged with their region colors. Regions with the same color are related together for feature fusion.

**Figure 9 sensors-20-02224-f009:**
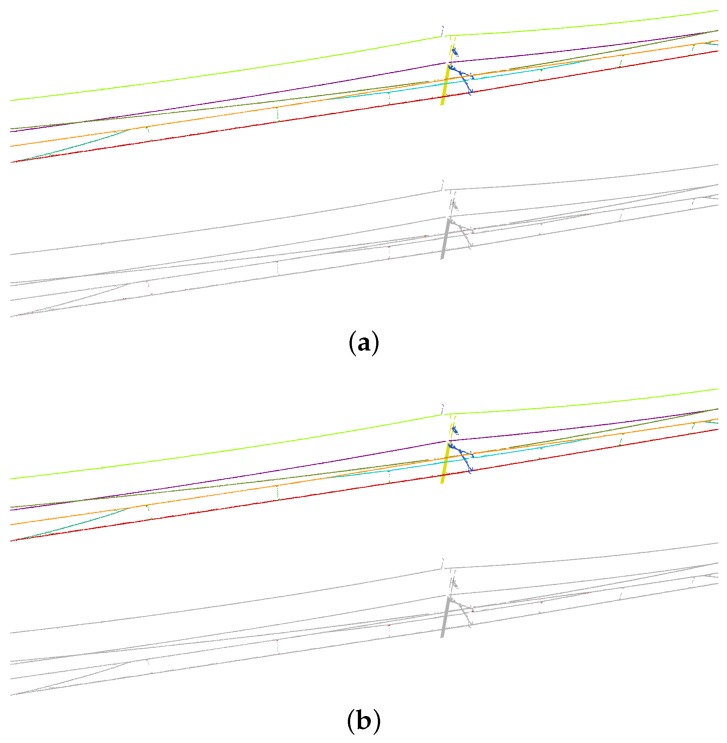
Segmentation results of non-insulated overlap section: (**a**) unidirectional mode; and (**b**) bidirectional mode. In each subfigure, predictions are shown above and errors are highlighted in red below (the same as Figure 10 and Figure 11).

**Figure 10 sensors-20-02224-f010:**
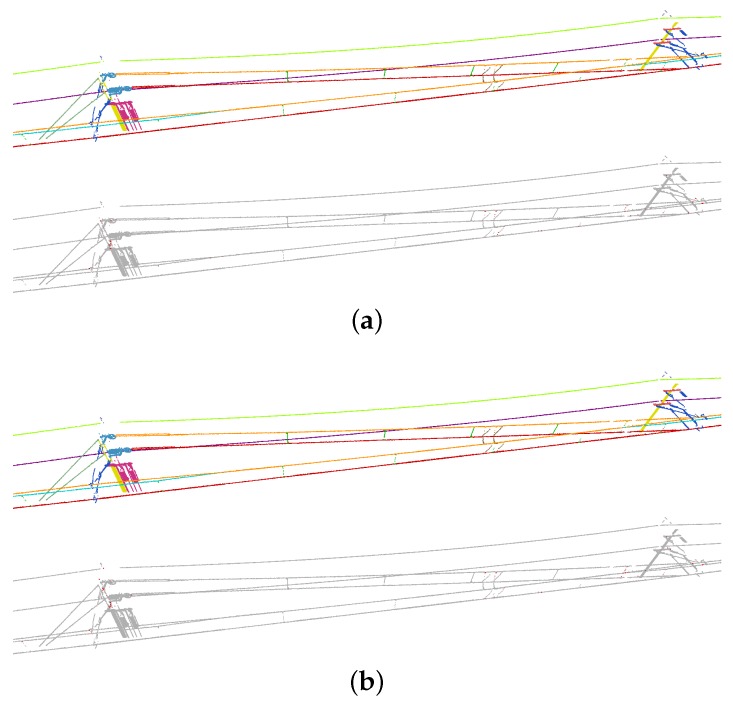
Segmentation results of insulated overlap section: (**a**) unidirectional mode; and (**b**) bidirectional mode.

**Figure 11 sensors-20-02224-f011:**
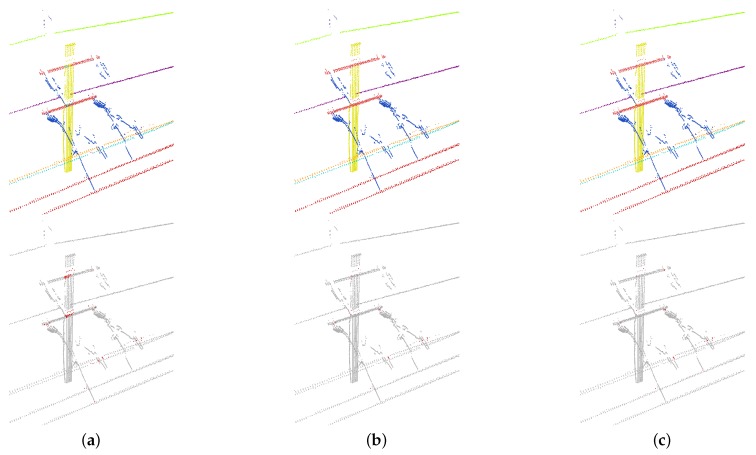
Segmentation results of a scene with spreaders: (**a**) non-feature fusion mode; (**b**) unidirectional mode; and (**c**) bidirectional mode.

**Table 1 sensors-20-02224-t001:** Parameters of our MLS system.

Parameters	Value
scaning frequency	25 Hz
angular resolution	$0.1667^{{}^{\circ}}$
field of view	$90^{{}^{\circ}}$
range	80 m
travelling speed	approximately 1 m/s

**Table 2 sensors-20-02224-t002:** The point proportions and object quantities of each class. Note that “-” in this table means that the class is uncountable.

Class	Training Dataset		Test Dataset	
	Points (%)	Quantity	Points (%)	Quantity
COW	17.90	-	4.18	-
CAW	12.75	-	2.84	-
CTLV	4.18	338	1.11	92
DRO	0.52	1982	0.12	536
SW	3.31	275	0.95	76
pole	8.05	278	2.15	76
SPD	0.61	124	0.15	32
RW	14.71	-	3.59	-
FW	13.31	-	3.29	-
SI	0.08	220	0.03	55
PB	0.69	34	0.13	8
TW	1.64	34	0.30	8
GW	0.78	108	0.24	26
MPAW	1.00	10	0.08	1
MPAAW	0.13	20	0.01	2
EC	0.1	29	0.02	8
others	0.94	-	0.12	-

**Table 3 sensors-20-02224-t003:** Per-class precision, recall, and IoU in the unidirectional mode and the bidirectional mode.

Class	Unidirectional Mode			Bidirectional Mode		
	Precision (%)	Recall (%)	IoU (%)	Precision (%)	Recall (%)	IoU (%)
COW	99.92	99.91	99.83	99.93	99.91	99.84
CAW	99.26	96.38	95.69	99.12	99.01	98.15
CTLV	99.39	99.19	98.59	99.36	99.21	98.59
DRO	96.50	93.90	90.80	97.04	92.91	90.35
SW	90.37	98.31	88.98	97.12	97.83	95.07
pole	99.61	99.68	99.29	99.55	99.76	99.32
SPD	97.20	98.86	96.13	98.24	97.77	96.08
RW	99.92	99.92	99.84	99.99	99.94	99.93
FW	99.99	99.99	99.98	99.99	99.98	99.97
SI	99.34	98.41	97.77	98.29	99.02	97.34
PB	97.89	99.41	97.32	99.06	99.25	98.32
TW	99.47	98.93	98.41	99.28	99.28	98.57
GW	98.93	97.72	96.70	99.18	99.12	98.31
MPAW	96.05	99.33	95.43	97.17	99.89	97.06
MPAAW	95.11	99.53	94.69	98.62	99.69	98.32
EC	95.35	90.59	86.75	96.77	90.90	88.23
others	98.41	99.36	97.79	98.94	99.56	98.51
mean	97.81	98.20	96.12	98.69	98.41	97.17

**Table 4 sensors-20-02224-t004:** Confusion matrix of partial classes in the unidirectional mode. Values denote the point proportion of the reference.

Reference	Prediction (%)						Total (%)
	COW	CAW	CTLW	DRO	SW	EC	
COW	99.91	0.00	0.05	0.03	0.00	0.00	99.99
CAW	0.00	96.38	0.04	0.05	3.41	0.01	99.89
CTLV	0.05	0.16	99.19	0.00	0.09	0.00	99.49
DRO	1.99	2.10	0.03	93.90	1.49	0.36	99.87
SW	0.01	1.58	0.04	0.04	98.31	0.00	99.98
EC	1.20	2.02	1.96	4.23	0.00	90.59	100

**Table 5 sensors-20-02224-t005:** Confusion matrix of partial classes in the bidirectional mode. Values denote the point proportion of the reference.

Reference	Prediction (%)						Total (%)
	COW	CAW	CTLW	DRO	SW	EC	
COW	99.91	0.00	0.05	0.03	0.00	0.00	99.99
CAW	0.00	99.01	0.05	0.03	0.83	0.01	99.93
CTLV	0.04	0.12	99.21	0.00	0.08	0.00	99.45
DRO	1.91	2.39	0.01	92.91	2.67	0.10	99.99
SW	0.01	2.12	0.03	0.01	97.83	0.00	100
EC	1.20	2.08	0.00	5.62	0.13	90.90	99.93

**Table 6 sensors-20-02224-t006:** Model size and Time. Value of model size denotes the total size of the model parameters. Value of time denotes the average time of processing per scan line on the test dataset. In the unidirectional mode and the bidirectional mode, scan lines are processed in batch. In the scan line-by-scan line (SLSL) unidirectional mode, scan lines are processed in sequence with the same model parameters as the unidirectional mode, which means that segmentation results of the unidirectional mode and SLSL unidirectional mode are the same.

Mode	Model Size (MB)	Time (ms)
unidirectional	1.11	1.80
bidirectional	1.74	1.95
unidirectional (SLSL)	1.11	8.99

**Table 7 sensors-20-02224-t007:** Comparison of different settings of region size threshold ε.

Settings of ε	Unidirectional Mode		Bidirectional Mode	
	mIoU (%)	OA (%)	mIoU (%)	OA (%)
0.5	84.29	97.80	88.73	98.35
2	77.73	96.32	82.47	95.96
7	87.99	96.95	90.24	97.85
0.5, 7	95.10	99.17	97.09	99.52
0.5, 2, 7	96.12	99.36	97.17	99.54

**Table 8 sensors-20-02224-t008:** Comparison of the results with multi-scale feature extraction (ε=0.5,2,7) in the non-feature fusion mode, the unidirectional mode, and the bidirectional mode.

Mode	mIoU (%)	OA (%)
non-feature fusion	93.53	98.29
unidirectional	96.12	99.36
bidirectional	97.17	99.54

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
