# Peer review of "A Deep Learning-Based Method for Overhead Contact System Component Recognition Using Mobile 2D LiDAR"

_sensors, 2020, doi:10.3390/s20082224_

Round 1

Reviewer 1 Report

The author combines the mobile laser scanning technology and the deep learning approach to generate the semantic point cloud where geometries of common OCS components can be extracted to promote the efficiency and automation of OCS inspection. However, this article is mainly about engineering application rather than theoretical innovation.

1. The abstract should be refined since it can not reflect the innovation of the theory manuscript.

2. According to lins105-106 in page3 in the related work, the theory contribuation of the manuscript is that the proposed deep learning-based method processes the data scan line by scan line rather than processes the whole point cloud, avoiding computing relations between points of different scan lines to reduce computational complexity. However, this contribuation is too small to be published as a regular research paper.

3. This article is mainly about engineering application rather than theoretical innovation. Thus, the contribuation of this manuscript should be refined.

4. The format of the references should be checked carefully.

Author Response

Point 1: 1. The abstract should be refined since it can not reflect the innovation of the theory manuscript.

 Response 1: We have modified the abstract to reflect the innovation and advancement of our method.

Point 2: According to lins105-106 in page3 in the related work, the theory contribution of the manuscript is that the proposed deep learning-based method processes the data scan line by scan line rather than processes the whole point cloud, avoiding computing relations between points of different scan lines to reduce computational complexity. However, this contribution is too small to be published as a regular research paper.

Response 2: This point is used to state roughly how our method process data. We move it into Section 3 because it might be more reasonable to serve as an explanation of the method framework.

Point 3: This article is mainly about engineering application rather than theoretical innovation. Thus, the contribution of this manuscript should be refined.

Response 3: To make the manuscript focus more on the theory rather than engineering solution, we have rewritten the contributions of this manuscript in the introduction and modified our conclusions.

Point 4: The format of the references should be checked carefully.

Response 4: We have checked the format of the references and correct the errors.

Furthermore, thank you for your kind advice and reminding.

Reviewer 2 Report

This manuscript explores the possibilities of applying mobile laser scanning (MLS) technology composed of a 2D LiDAR and an odometer sensor for overhead contact system (overhead cantenary system) inspections. These are necessary to guarantee the safety of a railway operation. Deep learning algorithms are applied to generate the semantic point cloud where geometries of common OCS components can be extracted.

The paper has a concrete and clear structure. A good overview of the field is given, but there might be some papers missing on the topic of applying machine learning in the field of overhead contact systems (OCS) inspection. Additionally, it may be beneficial and necessary to analyze the way in which existing commercial OCS inspection solutions work.

There is room for improvement when it comes to the explanation of how the experiment was conducted. For example, no validation dataset is mentioned, i.e. it is unclear how the hyperparameter tuning was done, or why the batch size = 1 etc.

Author Response

Point 1: There might be some papers missing on the topic of applying machine learning in the field of overhead contact systems (OCS) inspection.

 Response 1: We have tried our best to search the related work, but there are few studies applying machine learning to MLS-based OCS inspections. Therefore, we add some deep learning-based methods for image-based OCS inspections in Section 2 (Related Work) as supplements.

Point 2: It may be beneficial and necessary to analyze the way in which existing commercial OCS inspection solutions work.

Response 2: We have presented some existing commercial OCS inspection solutions in Section 2 (Related Work) and analyzed the application of image-based OCS inspections and MLS-based OCS inspections.

Point 3: There is room for improvement when it comes to the explanation of how the experiment was conducted. For example, no validation dataset is mentioned, i.e. it is unclear how the hyperparameter tuning was done, or why the batch size = 1 etc.

Response 3: We have added more details of how the deep learning-based modules are optimized in Section 4.2 (Implementation Details).

Furthermore, thank you for your kind advice.

Round 2

Reviewer 1 Report

The author has revised the manuscript. It is better than the previous version. However, this article is still mainly about an engineering application rather than theoretical innovation.

1. According to the authour, there are 3 contributions of the manuscript. However, only the proposed iterative point partitioning algorithm is the theoretical contribution. The other two factors are engineering applications rather than innovation. Thus, the introduction and abstract should be re-written againg to show the innovation, not just an engineering application.

2. The engineering application has its own characteristics in this manuscript. However, it is necessary to extract scientific problems from engineering applications. The drawback of this manuscript is that it fails to extract an suitable scientific problem. Therefore, the logic of the manuscript needs to be revised again.

3. All of the formular shoud be numbered in sequence.

4. The format of the references should be checked again and unified.
